# Deep Learning for Generalized EEG Seizure Detection after Hypoxia–Ischemia—Preclinical Validation

**DOI:** 10.3390/bioengineering11030217

**Published:** 2024-02-24

**Authors:** Hamid Abbasi, Joanne O. Davidson, Simerdeep K. Dhillon, Kelly Q. Zhou, Guido Wassink, Alistair J. Gunn, Laura Bennet

**Affiliations:** 1Department of Physiology, Faculty of Medical and Health Sciences, University of Auckland, Auckland 1023, New Zealand; joanne.davidson@auckland.ac.nz (J.O.D.); aj.gunn@auckland.ac.nz (A.J.G.); 2Auckland Bioengineering Institute (ABI), University of Auckland, Auckland 1010, New Zealand

**Keywords:** seizure detection, deep learning, neonatal hypoxic–ischemic encephalopathy, hypothermia, full-term and preterm fetal sheep, EEG, pattern recognition

## Abstract

Brain maturity and many clinical treatments such as therapeutic hypothermia (TH) can significantly influence the morphology of neonatal EEG seizures after hypoxia–ischemia (HI), and so there is a need for generalized automatic seizure identification. This study validates efficacy of advanced deep-learning pattern classifiers based on a convolutional neural network (CNN) for seizure detection after HI in fetal sheep and determines the effects of maturation and brain cooling on their accuracy. The cohorts included HI–normothermia term (*n* = 7), HI–hypothermia term (*n* = 14), sham–normothermia term (*n* = 5), and HI–normothermia preterm (*n* = 14) groups, with a total of >17,300 h of recordings. Algorithms were trained and tested using leave-one-out cross-validation and *k*-fold cross-validation approaches. The accuracy of the term-trained seizure detectors was consistently excellent for HI–normothermia preterm data (accuracy = 99.5%, area under curve (AUC) = 99.2%). Conversely, when the HI–normothermia preterm data were used in training, the performance on HI–normothermia term and HI–hypothermia term data fell (accuracy = 98.6%, AUC = 96.5% and accuracy = 96.9%, AUC = 89.6%, respectively). Findings suggest that HI–normothermia preterm seizures do not contain all the spectral features seen at term. Nevertheless, an average 5-fold cross-validated accuracy of 99.7% (AUC = 99.4%) was achieved from all seizure detectors. This significant advancement highlights the reliability of the proposed deep-learning algorithms in identifying clinically translatable post-HI stereotypic seizures in 256Hz recordings, regardless of maturity and with minimal impact from hypothermia.

## 1. Introduction

Perinatal hypoxic–ischemic encephalopathy (HIE) is a life-threatening complication, arising after lack of oxygen and blood flow to the brain [1]. In both experimental and clinical studies in term and preterm neonates, moderate to severe hypoxia–ischemia (HI) is typically followed by electroencephalographic (EEG) seizures, which are associated with white and grey matter injury [2,3,4,5]. While studies indicate a strong association between neonatal seizures and outcomes in term infants with HIE [6], there is evidence demonstrating that EEG seizures may not necessarily have clinical correlates in all neonates [7]. The great majority of EEG seizures at all ages are not associated with clinical signs. Real-time manual identification and interpretation of EEG seizures in newborns are, however, challenging and require expert clinical knowledge [8,9]. We need to significantly improve our ability to reliably detect seizures in order to more effectively target treatment to infants who might benefit [8,10].

Preclinical experiments can help to improve our understanding of the diagnostic and prognostic values of EEG as the timing and severity of the insult can be controlled and continuous measurements made from immediately after HI [11,12,13]. Neonatal models of clinically relevant HI insults in large animals at term are challenging and are largely carried out under anesthesia [14,15] and not readily feasible at preterm equivalent ages. A useful alterative is the fetal sheep, which permits continuous, comprehensive physiological measurements at all ages including continuous EEG recordings before, during, and after HI without the confounding effects of anesthesia or other medications. Fetal studies have shown that the phases of HI injury are similar to those seen in newborns, with reperfusion, latent, secondary, and tertiary phases of evolving injury.

We have previously demonstrated that EEG waveforms, specifically micro-scale sharp waves and gamma spike transients, are observed in the latent phase of recovery in preterm fetal sheep and are correlated with HI-related brain injury [16,17]. We have recently developed and validated successful data-driven deep-learning-based pattern classifiers that can accurately identify and quantify these transients [16] as well as delta-band rolling waveforms in the form of stereotypic evolving micro-scale seizures (SEMSs) in the EEG of preterm fetal sheep. The techniques utilized included 1D convolutional neural networks (1D-CNNs) [16,18], wavelet Fourier CNNs (WF-CNNs) [16], and a highly accurate state-of-the-art wavelet scalogram CNN (WS-CNN) approach that infuses spectrally rich feature maps of EEG sections into a deep CNN classifier for pattern recognition [16].

After initial recovery of oxidative metabolism in a “latent” phase after HI, moderate to severe injury is associated with secondary deterioration, with of loss of cerebral oxidative metabolism and high-amplitude stereotypic evolving seizures (HASs) in most fetuses and a fully stochastic EEG background regardless of age (Figure 1A–C) [11,19]. Experimentally, seizures typically start around 7–8 h post-HI and peak by approximately 24–48 h. In the preterm brain, these seizures are most often discrete events [11] but can develop into status epilepticus in term fetuses [19]. HASs are defined by their repetitive, stereotypic nature, lasting for at least 10 s with an amplitude of >20 µV in at least one EEG channel [20,21], matching the clinical definition [22,23,24]. Examples of HASs from HI–normothermia preterm, HI–normothermia term, and HI–hypothermia term groups are shown in Figure 1D–L. We have previously shown that the 1D-CNN and WS-CNN classifiers can be used to identify HASs after HI in preterm fetuses [18,25]. Others have shown that data-driven deep-learning-based classifiers can help to identify seizures in term and preterm infants [26,27] and adults [28,29].

While there is growing evidence that automated classifiers for EEG background assessment and grading are feasible [30,31,32,33], there is limited evidence for how automated deep-learning-based seizure detection algorithms perform across different stages of maturation [26]. Other studies also show the application of deep-learning-based algorithms for neonatal EEG seizure identification [34,35,36,37,38,39]. Recent studies emphasize that automated neonatal seizure detection algorithms can be beneficial; however, the algorithm’s predictions will still necessitate review by a human expert [5]. This is important to determine, as EEG characteristics change with age, consistent with maturation of neural connectivity and changing neurotransmitter function, and there are data to show that seizure morphology may also differ with age [11,26,40,41,42]. Further, there are few studies on the impact of clinical treatments, particularly therapeutic hypothermia (TH), which is the only currently approved neuroprotection treatment for term and near-term newborns with moderate to severe HIE [1]. Preclinical studies demonstrated that TH is most effective when started as early as possible within the first 6 h after HI (i.e., during the latent phase) and continued for 3 days. These data informed clinical trials and current practice [1]. Animal and clinical studies show that TH can partially suppress HASs [2,5,43,44,45,46].

In the present study, we sought to validate the accuracy of the WS-CNN, the WF-CNN, and 1D-CNN advanced deep-learning algorithms for seizure detection in a cohort of 40 near-term and preterm fetal sheep after severe HI, using continuous EEG recordings. In the near-term fetuses, we also examined whether TH affected their accuracy. We report that the proposed deep net algorithms can accurately detect seizures regardless of morphological variation due to age or hypothermia with a consistent decaying performance order when tested on data from HI–normothermia preterms, HI–normothermia terms, and HI–hypothermia terms, respectively. We will further demonstrate that training of the deep nets by adding data from sham–normothermia terms into the training sets can help to improve accuracy. The proposed spectral-based seizure detectors may also help to interpret spectral-component coherency of seizures. The non-denoised 256Hz recordings used in this study provided a framework to address the performance of the proposed deep-CNN-based seizure detectors in preclinical situations, where seizures could be reliably identified from non-HAS events (e.g., due to movement artifacts, etc.), thus making the validation of the techniques clinically trustworthy. These results are a significant step forward towards robust identification of post-HI seizures regardless of the brain maturity and treatment that should be further assessed in clinical studies. Finally, we show how these results illustrate a robust approach to developing a generalized seizure detector that can robustly identify seizures across all groups regardless of maturation or use of TH.

## 2. Methods

The known maturational differences between term and preterm EEG activity [47,48] imply that automatic identification of seizures across these groups can be challenging when the algorithm is trained only using data from one group (e.g., trained on terms and then tested on preterms). These developmental impacts have been emphasized by studies showing that adults epileptic seizure detectors are not suitable for EEG seizure detection in term infants [49,50]. Clinically, preterm seizure morphology and characteristics are more complex, with slightly different frequency components than those seen in term seizures [40,41]. In the current study, we investigate solutions to the challenges above by addressing the following questions:(a)Can our previous micro-scale EEG pattern classifiers be re-designed for accurate seizure identification in data from fetal sheep models with different gestational ages and/or under the influence of treatment with therapeutic hypothermia?

To answer this question, we refined previously developed EEG pattern classifiers, the WS-CNN, the WF-CNN, and the 1D-CNN, for seizure detection in preclinical data from a large cohort of fetal sheep. Figure 2 is a schematic showing where each algorithm is situated at a system level. We have described the basic algorithms previously [16,25]. Details of the re-designed structures including brief tables that demonstrate the new CNN architectures are described in Section 2.1, Section 2.2 and Section 2.3.

(b)Can the seizure detection algorithms trained/validated on datasets from certain group sets identify seizures in the EEG sets of other individual groups?

The paper explores how the choice of data partition (train/validation/test) influenced the results for each algorithm:Study #1: A leave-one-out cross-validation (LOOCV) approach where data from the term sham–normothermia group were included in three different training/test schemes.Study #2: A leave-one-out cross-validation (LOOCV) approach where data from the sham–normothermia group were excluded in three different training/test schemes. This was used to study the possible impacts of removing data from the sham–normothermia group.Study #3: A *k*-fold cross-validation (*k* = 5) approach where data from all groups were randomly combined and included in five different training/test schemes.

We explored how the performance of each seizure detector changed across and within each category above. The three proposed seizure detectors were trained/validated/tested over 1727 h of recordings in each separate study above (overall more than 17,300 h of recordings).

(c)Can specific training strategies help to improve the generalization and robustness of pattern classifiers to perform equally well across all groups and identify seizures regardless of what hemisphere the EEG has been recorded from?

Study #3 above showed that combining datasets recorded from both hemispheres of all four fetal sheep cohorts (*n* = 40) provided the most accurate generalized seizure detector. We then demonstrated that our state-of-the-art WS-CNN seizure classifier outperformed the other proposed algorithms for real-time identification of HASs and was able to distinguish them from high-amplitude EEG noise and/or other background activity in the conventional 256 Hz post-HI recordings, across all studies.

### 2.1. WS-CNN Seizure Detector

Deep-learning pattern recognition classifiers generally achieve optimally higher accuracies when fed with proper extracted features that provide higher-resolution information for decision making. We have previously shown that WS images, generated using a wavelet basis function with similar morphology/characteristics to the pattern of interest over a broad scale range, contain high-resolution spectral information and desirably provide superior feature maps for CNN classifiers [16]. In this study, we used the Morlet wavelet (see Figure 2 of [25]) as its morphological profile ideally matches the natural repetitive rhythms of high-amplitude stereotypic seizures.

The 2D scalogram images were generated by continuous wavelet transforming (CWT) the zero-meaned raw 256 Hz EEG time-series using the morl function over a large scale range between 1 and 500 (step size = 5). This scale range corresponds to a large pseudo-frequency range between 0.42 and 208 Hz (as EEGs were sampled at 256 Hz), ensuring that almost all of the spectral details of the EEG segments have been captured. Images were created at 500 × 333 pixels and stored as “.png” format with resolution (dots per inch) set at 300. Figure 3A–D and Figure 3E–H demonstrate examples of the HASs in the post-HI insult recordings and their corresponding WSs, respectively. Examples of non-HAS EEG segments and their corresponding WSs are shown in Figure 3I–L and Figure 3M–P, respectively. Figure 3E–H demonstrate how the spectral features of morl WSs of HAS segments, over a scale of 1:5:500, were visually distinguishable from the WSs of non-HAS EEG events shown in Figure 3M–P.

WS feature map images were then infused into a deep CNN classifier (WS-CNN) for pattern recognition [16]. Despite its larger architecture design and computationally intensive nature, this algorithm consistently outperforms our other CNN-based classifiers [16]. Table 1 details the architecture of the proposed 17-layer deep WS-CNN classifier, including convolutional layers (with batch normalization and ReLU units), max-pool, fully connected layers, and a softmax and a classification layer at the end (see full details in [16]). The root mean square propagation (RMSProp) optimizer was used to update the weights and bias parameters of the classifier. Due to the satisfactory overall performance from RMSProp, we did not investigate substituting this optimizer with Adam or SGDM optimizers. Training/validation was performed over 60 epochs using the default parameter values of 1.00 × 10^−3^ and 0.9 for the learning rate and SquaredGradientDecayFactor, respectively.

### 2.2. WF-CNN Seizure Detector

This approach is a simplified version of the WSs where only the spectrally dominant features of the raw EEG segment, in the form of wavelet and Fourier spectrums (WF), were extracted and used as opposed to the full-range spectral features (scalogram images) [16,25]. We combined three time-series to form 3D input matrix sets of 51,302 × 3 × 1 in size:-The CWT coefficients of each EEG segment using morl at an arbitrary scale of 80 (equal to pseudo-frequency of 2.56 Hz). This scale number was chosen to target the embedded spectrums near the mean frequency of the delta-band (0.5–4 Hz).-The inverse Fourier transform time-series of the EEG segment (IFFT: Spectral components within 0.2–4.5 Hz were preserved). This was chosen to cover delta-band spectrums as studies have shown neonatal seizures are more likely to contain rhythmic delta sharp-waves/discharges [41,51].-The original raw EEG segment.

The 3D input matrix sets of the spectrally dominant features from the previous section were fed into a deep 2D-CNN classifier (WF-CNN). We designed a 14-layer deep CNN to perform classification on the 3D input matrices of features. This approach is computationally more efficient due to the much simpler input features [16,25]. Table A1 of Appendix A details the architecture of the proposed 14-layer deep 2D-CNN used in the WF-CNN seizure detector. The inner convolutional layers of the network were designed to avoid massive size reductions within the inner layers. An RMSProp updating optimizer was used to train the WF-CNN over 60 epochs with default parameter settings.

### 2.3. 1D-CNN Seizure Detector

We recently demonstrated that a 1D-CNN classifier is capable of identifying HASs in a limited dataset of preterm fetal sheep [18]. This approach is computationally the simplest seizure classifier compared to the WF-CNN and WS-CNN approaches, where the feature extraction blocks in the two previous approaches were skipped and a CNN produces internal feature maps of its input 1D EEG time-series. In the current study, EEG segments of 51,302 × 1 in length were directly infused into a 14-layer deep 1D-CNN classifier. The designed architecture was trained using an RMSProp optimizer over 60 epochs with default parameter settings (Table A2 in Appendix A).

### 2.4. Performance Metrics

The performances of the proposed seizure detectors were evaluated by measuring sensitivity, selectivity, precision, accuracy, and area under the curve (AUC). These metrics were calculated by determining full results of the confusion matrix including the total number of true-positive (TP), true-negative (TN), false-positive (FP), and false-negative (FN) hits for each scheme in each study category, separately (see Table 2). Precision is the percentage of TPs among TP + FN. Accuracy is defined as the percentage of true hits (TP + TN) among all possible outcomes (TP + FN + TN + FP). Training-to-testing ratios/regimes in each scheme of study #1 and #2 were over-ruled by the number of EEG patterns in each animal group (Table 2). A fixed training-to-testing ratio of 4.0 was considered for all five cross-validation folds in study #3 (80% data for training/validation and 20% for test). Partitioning data across all folds of study #3 was performed using the “rng(‘default’)” Matlab function [52] to ensure data reproducibility for performance assessment across all seizure detectors.

### 2.5. Computing Infrastructure

The algorithms were developed, trained, and tested in Matlab^®®^ using New Zealand eScience Infrastructure (NeSI) (Auckland, New Zealand) high-performance computing facilities’ Cray CS400 cluster [53]. The training process was executed using enhanced NVIDIA Tesla A100 PCIe GPUs with 40 GB HBM2 stacked memory bandwidth at 1555 GB/s [54]. Intel Xeon Broadwell CPUs (E5-2695v4, 2.1 GHz) were used on the cluster for handling the GPU jobs.

### 2.6. Experiments, Data Acquisition, and Preparation

#### Ethics

All procedures were approved by the Animal Ethics Committee of the University of Auckland (R1942) under the New Zealand Animal Welfare Act and carried out in accordance with the Code of Animal Ethical Conduct established by the Ministry of Primary Industries of the New Zealand Government.

### 2.7. Surgical and Experimental Procedures

#### 2.7.1. Fetal Surgery

In brief, 26 near-term and 14 preterm time-mated Romney/Suffolk fetal sheep at 124 and 98 ± 1 days of gestation, respectively (term is 145 days of gestation), were used in this study. Fetuses were surgically instrumented with catheters and electrodes under general anesthesia using aseptic techniques, as previously described [11]. Prior to the surgical procedure, food was withheld from the animals for 18 h; however, access to water was maintained. Long-acting oxytetracycline (20 mg/kg, Phoenix Pharm, Auckland, New Zealand) was administered intramuscularly to the ewes 30 min before the commencement of surgery. Anesthesia was initiated through intravenous injection of propofol (5 mg/kg, AstraZeneca Limited, Auckland, New Zealand) and was sustained using 2–3% isoflurane in an oxygen environment. Throughout the procedure, the depth of anesthesia, maternal heart rate, and respiration were continuously monitored by experienced anesthetic personnel. The ewes received a continuous infusion of isotonic saline (at an approximate rate of 250 mL/h) to uphold fluid balance. Following a maternal midline abdominal incision, the fetus was exposed. For near-term fetuses, the vertebral–occipital anastomoses were ligated and inflatable carotid occluder cuffs were placed around both carotid arteries [55]. For preterm fetuses, an inflatable silicone occluder (OC16HD, 16 mm, In Vivo Metric, Healdsburg, CA, USA) was loosely positioned around the umbilical cord to enable postsurgical occlusion of the umbilical cord for inducing fetal HI [11]. Using a 7-stranded stainless steel wire (AS633–7SSF; Cooner Wire Co., Chatsworth, CA, USA), two pairs of EEG electrodes were constructed and placed on the dura mater over the parasagittal parietal cortex (10 mm and 20 mm anterior to bregma and 10 mm lateral for near-term fetuses and 5 mm and 15 mm anterior to bregma and 5 mm lateral for preterms) and secured with cyanoacrylate glue. A reference electrode was sewn over the occiput. For near-term fetuses, a thermistor (Replacement Parts Industries, Inc., Chatsworth, CA, USA) was placed over the parasagittal dura, 30 mm anterior to bregma, to measure extradural temperature and a second thermistor was inserted into the esophagus to measure core temperature. In the case of all fetuses, a cooling cap made from silicone tubing (3 × 6 mm, Degania Silicone, Degania Bet, Israel) was affixed to the head. For all fetuses, the uterus was then closed and antibiotics (80 mg Gentamicin, Pharmacia and Upjohn, Rydalmere, New South Wales, Australia) were administered into the amniotic sac. The maternal laparotomy skin incision was sutured and anesthetized with an injection of 10 mL of 0.5% bupivacaine plus adrenaline (AstraZeneca Ltd., Auckland, New Zealand). All fetal catheters and leads were externalized through an incision in the maternal flank. The maternal long saphenous vein was catheterized to provide access for postoperative maternal care and euthanasia.

Postoperative care: Following surgery, sheep were housed in individual metabolic cages with unrestricted access to food and water. The room was maintained at a temperature of 16 ± 1 °C with 50 ± 10% humidity and a 12 h light/dark cycle (lights on at 06:00 h). The ewe received daily intravenous antibiotics for four days (600 mg benzylpenicillin sodium, Novartis Ltd., Auckland, New Zealand, and 80 mg gentamicin).

The patency of fetal catheters was sustained by continuous infusion of heparinized saline (20 U/mL at 0.2 mL/h), and the maternal catheter was maintained by daily flushing.

#### 2.7.2. Experimental Protocols

Near-term fetuses were randomized to HI–normothermia term (HI, no hypothermia) (*n* = 7), HI–hypothermia term (HI, hypothermia) (*n* = 14), and sham–normothermia term (no HI, no hypothermia) (*n* = 5). Separately, an HI–normothermia preterm group was prepared (HI, no hypothermia, *n* = 14). For near-term fetuses, at 128 ± 1 d gestation, cerebral ischemia was induced through the temporary inflation of the carotid occluder cuffs with sterile saline for a duration of 30 min. The successful occlusion was then confirmed by the rapid onset of an isoelectric EEG signal, typically within 30 s after inflation. In the sham–normothermia experiments, the carotid occluder cuffs were not inflated. HI–normothermia preterm fetuses at 103 ± 1 d gestation received complete umbilical cord occlusion induced by inflation of the umbilical cord occluder with sterile saline. This occlusion was maintained for 25 min or until blood pressure dropped below 8 mmHg or asystole occurred.

In the HI–hypothermia term group, hypothermia was started 3 h after the end of the HI insult and continued for 72 h. Hypothermia was performed by attaching the exteriorized ends of the silicone scalp coil to a pump (TX150 heating circulator, Grant Instruments Ltd., Cambridge, UK) in a cooled water bath and circulating cold water through the cooling coil. In line with previous studies conducted on near-term fetal sheep, the initial target extradural temperature was regulated to a range of 31–33 °C in the HI–hypothermia terms [55]. The water was not circulated, and the cooling coil was maintained in thermal equilibrium with the fetal temperature. As described previously, once the cooling period concluded, the water pump was deactivated, and the fetuses were allowed to naturally re-warm over a period of approximately 60 min. Subsequently, euthanasia was administered to both ewes and fetuses at the conclusion of the study, which occurred 7 days after ischemia for near-terms or 21 days after ischemia for preterms. The euthanasia process involved an intravenous overdose of sodium pentobarbitone (9 g administered to the ewe; Pentobarb 300; Chemstock, Christchurch, New Zealand) for subsequent immunohistochemistry analysis. In all groups, 7 days of EEG recordings after HI were considered for seizure analysis. For the HI–hypothermia term group, this included seizures during the entire cooling phase and those that occurred after the end of cooling.

### 2.8. Data Acquisition

After the end of surgery, fetal EEG and other physiological parameters including mean arterial blood pressure, heart rate, and EEG were recorded continuously from 24 h before the insult and evaluated up to 168 h after the end of HI or sham HI.

Data were continuously acquired and stored on disk for subsequent offline analysis utilizing custom data acquisition software (LabView 2020 for Windows, National Instruments, Austin, TX, USA). The raw EEG recordings underwent initial processing, commencing with a 6th-order anti-aliasing Butterworth low-pass filter featuring a 500 Hz cut-off frequency. Following this step, the signal was subjected to a gain amplification of ×10,000 and subsequently high-pass filtered via a first-order filter, where the cut-off frequency was set at 1.6 Hz. The recordings were then digitized at a rate of 4096 Hz, and a 10th-order low-pass inverse Chebyshev filter at 128 Hz (implemented in software) was applied, subsequently reducing the sample rate to 256 Hz before the data were saved. The data from this final stage were subsequently transferred to Matlab for analysis.

EEG intensity (power) was also derived using the sum of the power spectrum between 1 and 20 Hz over 1 min recording bins of the raw EEG signal and log transformed (decibels (dB), 20× log (intensity)) as this transformation gives a better approximation of the normal distribution.

A total of ~10,080 min of EEG recordings were collected from each fetal sheep (in total more than 8600 h per EEG channel from 40 subjects). We manually identified the location of all HAS events on both left and right EEG channels by plotting the EEG power (log μv^2^), separately for each, and the locations of the seizures were marked accordingly. The exact seizure section was then identified in the raw EEG recording and the seizure was centered in a 51,302 × 1 long segment (equal to ~3.34 min long segments). The procedure of centralizing the seizure has been explained in the preprocessing section below. Choosing the 3 min long segments ensured that the full lengths of the longer seizures were included in the dataset. A total of 31,015 EEG segments including 3955 HASs and 27,060 non-HASs were manually annotated to create the database (Table 3). The larger number of non-HAS events was intentionally chosen to account for the significantly higher proportion of normal EEG activity and to encompass the natural emergence of seizures against the EEG background. This reinforces the algorithm’s robustness in facing non-HAS events. Table 3 provides full details for the number of EEG segments in each animal group. Table 2 further demonstrates how data were partitioned into different training schemes for each study category.

#### Preprocessing

The EEG power signal mainly helps to monitor the long-term trend of an experiment and is calculated as the sum of the power spectrum of the raw EEG signal between 1 and 20 Hz over a 1 min recording. Therefore, manually identified/labeled seizure points in the EEG power signal represent only 1 min data where the center/minimum/maximum datapoints are not necessarily the center of the seizure. Therefore, we developed a strategy to find and locate seizures in the center of the 3 min long EEG segment as follows:

We initially selected 1 min long recordings either side of the marked seizure location in the EEG power signal (sig_1). The max/min values of sig_1 are not necessarily a good choice for the center of a seizure, thus we chose to consider the center of the weight of sig_1 (the signal’s datapoint with the highest weight) as the center of the seizure epoch. To find the center of the weight, sig_1 was initially zero meaned and amplified to the arbitrary power of 10 to intensify signal components with higher energy levels. The output signal (sig_2) was then passed through a moving median absolute deviation (movmad) function with an arbitrary sliding window size of a length of 8000 points and scaled up to sig_1 (sig_3). This strategy allowed determination of sections of sig_3 that held the highest signal weight within the arbitrarily assigned moving average datapoints above (i.e., 8000). The maximum value of sig_3 was then considered as the center of weight for the identified seizure and 100 s of data from either side of this datapoint (equal to a total of 3.34 min) were automatically selected to form the seizure segment (HAS). This step was initially performed by developing automatic algorithms to generate plots, similar to what is shown in Figure 4, for all manually identified seizure locations.

The plotted graphics were then visually checked by two experts to ensure that the final EEG segment represented a seizure that is centered in the 3 min long EEG segment. This preprocessing step can significantly improve the results of the proposed seizure classifiers as this centralizing strategy helps to use the data in a much better format for a classification task.

The non-HAS EEG segments were randomly chosen outside the HAS intervals (or overlapping tails of seizures in their segments). Therefore, the non-HAS events could include any electrophysiological activity including normal increasing EEG activity, background EEG, movement artifact, electronic noise, etc.

To improve generalization and strengthen the performance validity of the proposed pattern classifier, we did not de-noise the original EEG segments. This helps facilitate assessment of the proposed seizure detector on data that are similar to those obtained clinically. No data augmentation (such as horizontal/vertical flipping of the EEG segments) was performed on the data so that the performance assessments were purely evaluated on the natural electroencephalographic morphology of the patterns.

## 3. Results

### 3.1. Results of the WS-CNN Seizure Detector

Results of the WS-CNN approach for the three different study categories are shown in Table 4. The deep WS-CNN classifier was able to identify seizures in an unseen fetal sheep group regardless of age or treatment with an average high accuracy of 98.52% whether EEG data from the sham–normothermia group were used in the training set or not. The overall accuracy of the WS-CNN classifier for study #1 was equal to 98.5 ± 1.0% (range 97.2–99.7%), for study #2 98.6 ± 0.9% (range 97.4–99.7%), and for study #3 99.8 ± 0.04% (range 99.3–99.6%).

The results confirmed the reliability of the 17-layer deep WS-CNN pattern classifier for detection of post-HI HASs in the 256 Hz sampled EEG. Average area under curve (AUC) values of 0.95, 0.95, and 0.99 were obtained for studies #1 to #3, respectively. Figure 5A–C show the ROC curves and the corresponding AUC values of the WS-CNN seizure detector for the three study categories.

### 3.2. Results of the WF-CNN Seizure Detector

The confusion matrix results of the WF-CNN approach for the three different study categories are tabulated in Table 5. The overall accuracy of the WF-CNN approach for study #1 was equal to 98.5 ± 1.3% (range 96.8–99.8%), for study #2 98.1 ± 1.4% (range 96.4–99.7%), and for study #3 99.7 ± 0.08% (range 98.8–99.6%). 

The results showed that the 14-layer WF-CNN pattern classifier can also accurately identify the post-HI HASs. Average AUC values of 0.95, 0.95, and 0.99 were obtained for studies #1 to #3, respectively. Figure 5D–F show the ROC curves and the corresponding AUC values of the WF-CNN seizure detector for the three study categories.

### 3.3. Results of the 1D-CNN Seizure Detector

Confusion matrix results of the 1D-CNN classifier for each of the study categories are given in Table 6. The 14-layer deep 1D-CNN seizure detector was able to identify HAS/non-HAS patterns with an overall accuracy of 98.0 ± 1.0% (range 96.7–99.0%) for study #1, 97.6 ± 0.9% (range 97.0–98.9%) for study #2, and 99.7 ± 0.1% (range 99.5–99.9%) for study #3. 

The results indicated that despite the negligible fall in accuracy with the 1D-CNN approach, it could also reliably accurately identify HASs. Average AUC values of 0.95, 0.94, and 0.99 were obtained by the 1D-CNN approach for studies #1 to #3, respectively. Figure 5G–I show the ROC curves and the corresponding AUC values of the 1D-CNN seizure detector for the three study categories.

## 4. Discussion

The present study demonstrates that deep-learning pattern classifiers that we originally developed to identify EEG transients can accurately identify post-HI seizures with more than 99.7% accuracy across preterm and near-term gestations and during TH in near-term animals. In these studies, we examined whether data from different gestational ages could be used to train deep net classifiers to identify seizures from non-HAS EEG segments in unseen data from other groups. We explored three main study categories to test and compare algorithms using leave-one-out cross-validation approaches with training sets that (1) included and (2) excluded data from the sham–normothermia group; and (3) a *k*-fold cross-validation (*k =* 5) approach where data from all groups were randomly combined and included in the training set. We found some variations in classification accuracy across groups that likely reflect embedded morphological differences in seizures between groups.

Study #1 (sham–normothermia group included in the training sets):

The seizure detectors tested in this study consistently achieved their best performance when tested on data from HI–normothermia preterms (Figure 5A,D,G and Table 7) with an average accuracy of 99.5 ± 0.3%. This suggests that the embedded spectral morphological features of seizures in the HI–normothermia term and the HI–hypothermia term groups are also present in the preterm group.

Interestingly, when we included the preterm datasets in the training sets this reduced the average overall performances to 98.7 ± 0.2% and 96.9 ± 0.3% when the seizure detectors were tested on data from the HI–normothermia terms and HI–hypothermia terms, respectively. This indicates that the term datasets include more complex spectral features than the preterm data, and thus, while term-trained algorithms accurately identify HI–normothermia preterm seizures, there is less accuracy the other way around. This could be due to slower delta activity in the preterm fetuses (equivalent to a subset of delta-band full spectral range) compared to the evolving EEG components in a more term brain that covers a broader spectral delta range including those that already exist in preterms. This is evident from the increasing number of wrongly detected seizures (false positives) in study #1 in Table 4, Table 5 and Table 6 for the HI–sham–term–tested and HI–hypothermia–term–tested classifiers compared to the HI–normothermia–preterm–tested classifiers.

Our data are consistent with the well-known relative maturation and connectivity of the preterm brain compared with term. Our HI–normothermia preterm fetal group is at a brain age equivalent of 28–30 weeks of gestation in humans [11]. This age is characterized by rapid production of white matter, but at a time when the cortical neurons are not yet myelinated [56,57,58,59]. The EEG is discontinuous in nature, and the frequency content of preterm EEG generally contains slower delta activity [60,61], which develops into covering faster spectral components towards term age with the shift to continuous EEG activity and sleep state cycling [60,62,63,64,65].

Study #2 (sham–normothermia term group excluded from the training sets):

This study showed a similar profile to that of study #1, with data demonstrating decreasing performance from HI–normothermia preterms to HI–normothermia terms and HI–hypothermia terms. The lack of EEG data from the sham–normothermia term group in the training sets appeared to have contributed to lowering the overall performances of the proposed seizure detectors in this study category. The performance of the 1D-CNN classifier was impacted more than that of the WF-CNN and the WS-CNN classifiers, confirming the robustness of the latter two strategies.

Compared to study #1, adding data from the sham group in the training sets could also assist classifiers to achieve better performances by lowering the number of FN hits, on average. 

Study #3: The cross-validation results in Table 4, Table 5 and Table 6 and the ROC curves in Figure 5C,F,I indicate that the three proposed seizure detectors were able to properly generalize when trained on combined datasets from all groups and accurately identify seizures from non-seizures and background EEG activity. Results indicate how the seizure detectors perform equally well across all validation folds with average overall performances of 99.78 ± 0.04%, 99.73 ± 0.08%, and 99.70 ± 0.14% for the WS-CNN, WF-CNN, and the 1D-CNN classifiers, respectively. These high-performance measures have been cross-validated over 31,015 EEG patterns and confirm that the proposed classifiers are able to learn the morphological variations of seizures (and their possible spectral feature differences) across all fetal sheep groups to properly identify a novel HAS event in an unseen dataset.

These promising results indicate an effective approach to train generalized seizure detectors to robustly identify seizures regardless of the gestational age and/or the influence of treatment/drug on the EEG quality. The results also suggest that the simpler architectures of the 1D-CNN and the WF-CNN classifiers can be used to achieve much faster analysis, in real-time, instead of the computationally heavy structure of the WS-CNN. Nevertheless, the larger standard deviations of 0.14 and 0.08 from the 1D-CNN and WF-CNN, respectively, compared to 0.04 with the WS-CNN classifier illustrate a potential limitation of the faster classifiers. 

Algorithm comparisons: Overall, comparing performances of the classifiers in Table 4, Table 5 and Table 6 demonstrates that feeding the deep 2D-CNN classifiers with the spectrally rich feature maps of the EEG segments can provide much better seizure detection accuracies compared to when the raw EEG segments were directly fed into a 1D-CNN pattern classifier. In fact, the wavelet scalograms in WS-CNNs and the matrices of spectrally dominant features in the WF-CNNs provided robust spectrally detailed inputs for the deep 2D-CNN pattern classifiers to desirably classify post-HI HAS from non-HAS events. This unique ability to almost perfectly identify HASs from non-HASs is important as the non-HAS events in this study are any non-HAS electrophysiological activity that could include normal increasing EEG activity, movement artifacts, electronic noise, etc. Our data show that the proposed CNN-based seizure detectors can competitively identify the HASs across all study schemes with negligible performance drop for the WF-CNN and 1D-CNN compared to the WS-CNN approach. Data further suggest that the raw EEG time-series fed into the 1D-CNN as well as the spectrally dominant features in the input matrices of the WF-CNN approach can provide sufficient information for the designed CNN classifiers to identify HAS from non-HAS events. This is particularly important as the structure of the 1D-CNN and WF-CNN seizure detectors are computationally more efficient, allowing the algorithms to run faster with less required memory compared to the WS-CNN. The choice of faster computations and technology requirements is at the expense of negligible accuracy drop, compared to the WS-CNN strategy. The data suggest that the WS generated using a morl mother wavelet of a scale of 1–500 covered a broader range of spectral features that helped the WS-CNN to outperform the WF-CNN and 1D-CNN approaches. The application of the NVIDIA A100 GPUs in this study has contributed to remarkably fast analysis despite the computationally intensive nature of the processing.

Limitations: In interpreting this study, the reader should consider that, regarding the clarity in identifying HASs in dural surface-recorded EEGs from fetal sheep models, they have lower background noise than typical neonatal scalp recordings [66,67,68]. While the HASs observed in dural surface-recorded EEG from fetal sheep models are directly comparable to typical neonatal seizures in clinical scalp measurements, the dural recordings inherently provide a superior signal-to-noise ratio than conventional clinical scalp measurements. The enhanced clarity is beneficial for accurate seizure detection in this experimental setting but might not fully capture the challenges associated with clinical EEG recordings from the scalp, where noise levels are typically higher due to factors such as infant movement. Further, it is possible that movement may differ between the term and preterm brain. Thus, validation using clinical recordings will be an important future direction.

Nevertheless, these results support that the proposed algorithms can reliably identify post-HI seizures in conventional 256Hz recordings across groups with different gestational ages and/or under the influence of therapeutic hypothermia.

## 5. Conclusions

This study demonstrated the effectiveness of a deep CNN pattern classifier for generalized seizure detection in over 17,300 h of EEG recordings following acute HI in a cohort of 40 fetal sheep. The cohort included a range of settings, including normothermia vs. hypothermia, and term and preterm gestations, as well as sham controls. The CNN seizure classifier exhibited exceptional accuracy, with an average 5-fold cross-validated performance exceeding 99.7%, affirming the reliability of the proposed deep-learning algorithm. Minimal influences of gestational age and hypothermic treatment on seizure detector performance were observed, confirming its generalizability to identify seizures across gestational ages when trained on a robust dataset with samples from all groups. The three study categories, incorporating variations in training datasets, provided valuable insights into the robustness and generalization capabilities of the proposed algorithms. The classifiers consistently performed well across different validation folds, confirming their ability to identify seizures regardless of gestational age or therapeutic intervention. Notably, when trained on term data, the algorithms demonstrated superior accuracy in identifying seizures in preterm subjects compared to the reverse scenario, emphasizing the importance of considering the developmental stage of the fetal brain in training seizure detection algorithms. Variations in classification accuracy across study categories underscored morphological differences in seizures between groups, reflecting the relative maturation and connectivity of preterm versus term brains.

The study introduced the potential for real-time analysis using simpler architectures like the 1D-CNN and WF-CNN classifiers, offering faster processing while maintaining competitive accuracy, emphasizing the feasibility of deploying these algorithms in near-clinical scenarios. The comparative analysis of different CNN architectures highlighted the advantage of utilizing spectrally rich feature maps in 2D-CNN classifiers for improved seizure detection accuracy. The use of non-denoised 256Hz recordings added clinical relevance, showcasing the algorithms’ robustness in reliably identifying post-HI seizures under real-world conditions. Critical future directions will involve validating the algorithms with clinical data to establish their applicability in real-world healthcare settings. Overall, this research signified a significant advancement in automated seizure detection after HI events, offering a promising avenue for enhancing clinical practices and patient outcomes in neonatal care.

## Figures and Tables

**Figure 1 bioengineering-11-00217-f001:**
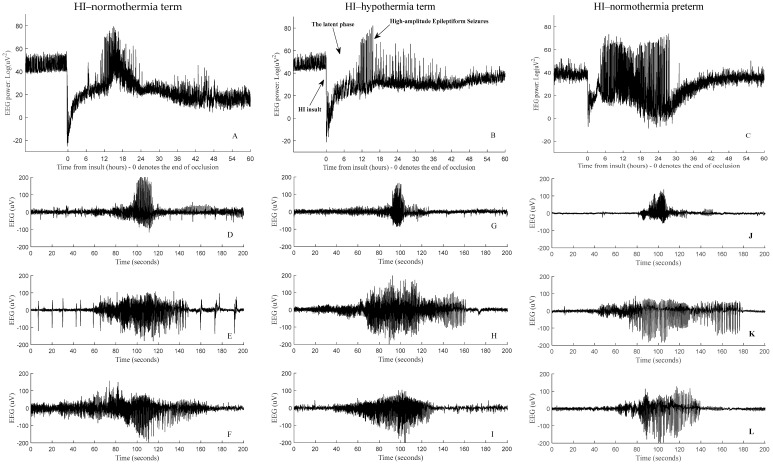
Examples of EEG power activity before, during, and up to 60 h after an HI insult induced by acute umbilical cord occlusion in an HI−normothermia preterm fetus (**A**), global cerebral ischemia induced by bilateral carotid artery occlusion in HI−normothermia term fetuses with no treatment (**B**), and HI−hypothermia term fetal sheep with 3 days of therapeutic hypothermia (**C**). The presence of high-amplitude stereotypic evolving seizures (HASs) during the secondary phase of recovery is demonstrated in (**A**–**C**). Examples of individual HASs from each fetal sheep in the above groups are shown in (**D**–**L**), respectively. HAS patterns above demonstrate examples of EEG epochs used in the training and testing of the proposed seizure detectors.

**Figure 2 bioengineering-11-00217-f002:**
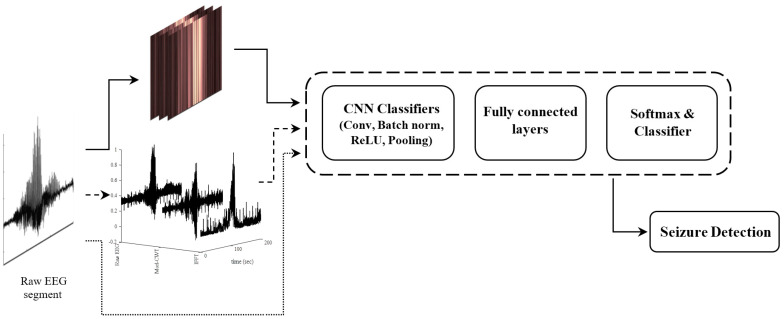
The architectures of the proposed seizure detectors. WS−CNN (solid line), WF−CNN (dashed line), 1D−CNN (dotted line).

**Figure 3 bioengineering-11-00217-f003:**
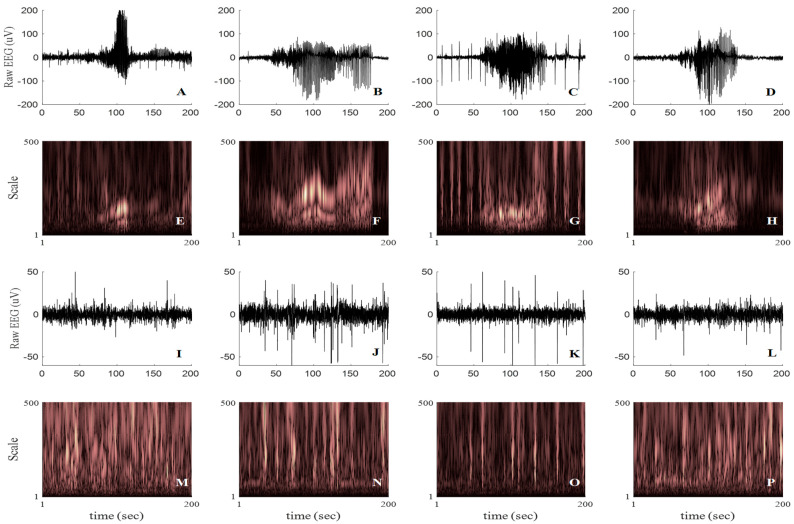
(**A**–**D**): Examples of post-HI high-amplitude stereotypic seizures (HASs) in all fetal sheep groups, equal to Figure 1D,E,K,L. (**I**–**L**): Examples of post-HI non−HAS EEG segments. (**E**–**H**) and (**M**–**P**): Examples of the corresponding scalogram images of the HAS and non−HAS EEG segments, respectively, used for training of the seizure detectors. The scalograms were generated using Morlet mother wavelet of scales 1 to 500.

**Figure 4 bioengineering-11-00217-f004:**
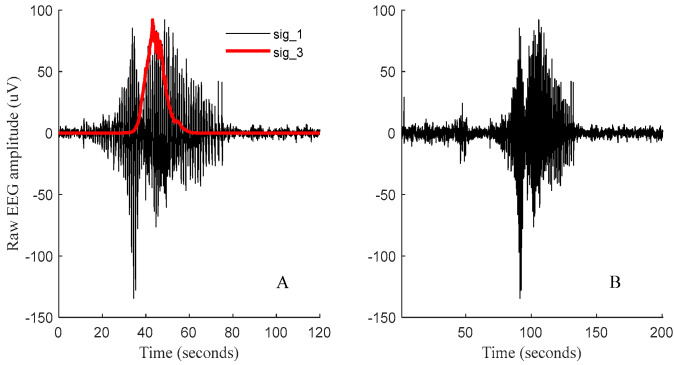
(**A**): sig_1: An example of un−-centered seizure (HAS) from preterm fetal sheep (10 h post−HI insult). sig_3: evaluated moving median absolute deviation (movmad) of sig_1 with an arbitrary sliding window size of length of 8000 points, scaled up to sig_1 for visualization. (**B**): Example of the final centered seizure (HAS) in a 3.34 min long EEG segment used in the training and testing of the deep net classifiers.

**Figure 5 bioengineering-11-00217-f005:**
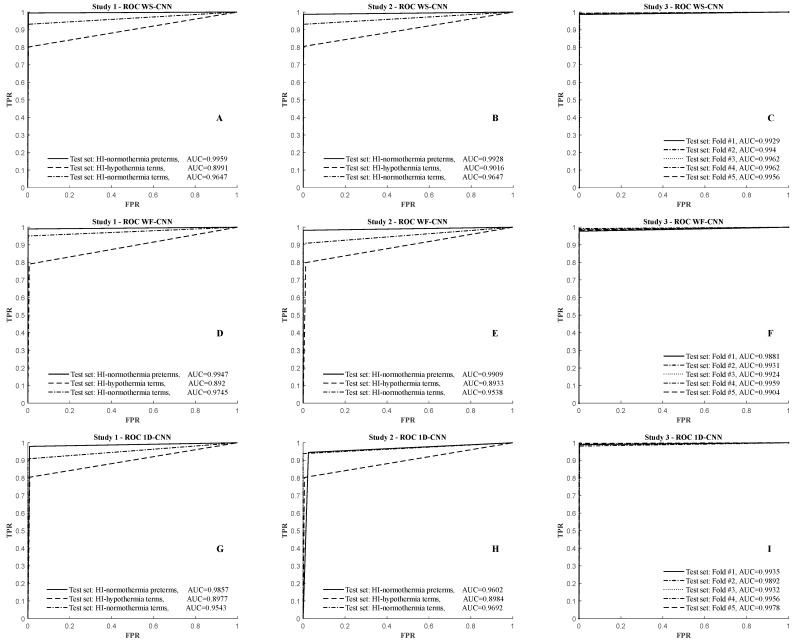
ROC curves and the corresponding AUC values from testing WS−CNN (**A**–**C**), WF−CNN (**D**–**F**), and 1D−CNN (**G**–**I**) seizure detectors in study #1, #2, and #3, respectively. The data for each proposed classifier are presented as mean ± SD, demonstrating improved accuracy and much lower variability when data from all fetal sheep groups have been used in the cross-validation results of study #3.

**Table 1 bioengineering-11-00217-t001:** The architecture of the proposed 17-layer deep WS-CNN seizure classifier.

Layers	Type	No. of Neurons (Output Layer)	Kernel Size	Stride	Padding	No. of Filters
0–1	Conv.	333 × 500	3	1	1	32
1–2	Max_pool	166 × 250	[3 2]	2	0	
2–3	Conv.	166 × 250	3	1	1	48
3–4	Max_pool	83 × 125	2	2	0	
4–5	Conv.	83 × 125	3	1	1	64
5–6	Max_pool	41 × 62	3	2	0	
6–7	Conv.	41 × 62	3	1	1	96
7–8	Max_pool	20 × 31	[3 2]	2	0	
8–9	Conv.	20 × 31	3	1	1	128
9–10	Max_pool	10 × 15	[2 3]	2	0	
10–11	Conv.	10 × 15	3	1	1	192
11–12	Max_pool	4 × 6	[4 5]	2	0	
12–13	Conv.	4 × 6	3	1	1	256
13–14	Max_pool	2 × 3	2	2	0	
14–17	Fully_connected	1536				
	Fully_connected	24				
	Fully_connected	2				
Output	Softmax and Classification					

**Table 2 bioengineering-11-00217-t002:** Study design including three main categories. Evaluations performed separately for the WS-CNN, WF-CNN, and 1D-CNN seizure detectors.

	Trained/Validated on	No. of EEG Patterns in the Training Set (Total/Seizures/Non_Seizures)	Total Length of Training Set (in Hours/Days)	Test on	No. of EEG Patterns in the Testing Set (Total/Seizures/Non_Seizures)	Total Length of Testing Set (in Hours/Days)	Training-to-Testing Ratio
**Study #1** 3 schemes	G1 + G2 + G3	20,491/2311/18,180	1024.6/42.7	G4	10,524/1644/8880	526.2/21.9	1.95
	G1 + G4 + G3	20,852/2652/18,200	1042.6/43.4	G2	10,163/1303/8860	508.2/21.2	2.05
	G2 + G4 + G3	25,582/2947/22,635	1279.1/53.3	G1	5433 /1008/4425	271.7/11.3	4.71
**Study #2** 3 schemes	G1 + G2	15,596/2311/13,285	779.8/32.5	G4	10,524/1644/8880	526.2/21.9	1.48
	G1 + G4	15,957/2652/13305	797.9/33.2	G2	10,163/1303/8860	508.2/21.2	1.57
	G2 + G4	20,687/2947/17,740	1034.4/43.1	G1	5433/1008/4425	271.7/11.3	3.81
**Study #3** 5 folds	G1 + G2 + G3 + G4 (Fold #1)	24,812/3164/21,648	1040.6/43.4	Fold #1 test-set	6203/791/5412	310.2/12.9	4.00
	G1 + G2 + G3 + G4 (Fold #2)	24,812/3164/21,648	1040.6/43.4	Fold #2 test-set	6203/791/5412	310.2/12.9	4.00
	G1 + G2 + G3 + G4 (Fold #3)	24,812/3164/21,648	1040.6/43.4	Fold #3 test-set	6203/791/5412	310.2/12.9	4.00
	G1 + G2 + G3 + G4 (Fold #4)	24,812/3164/21,648	1040.6/43.4	Fold #4 test-set	6203/791/5412	310.2/12.9	4.00
	G1 + G2 + G3 + G4 (Fold #5)	24,812/3164/21,648	1040.6/43.4	Fold #5 test-set	6203/791/5412	310.2/12.9	4.00

G1: HI–normothermia terms (*n* = 7); G2: HI–hypothermia terms (*n* = 14); G3: Sham–normothermia term control (*n* = 5); G4: HI–normothermia preterms (*n* = 14).

**Table 3 bioengineering-11-00217-t003:** Number of fetal sheep in each group as well as the total number of manually identified seizures and non-seizure patterns from the left/right EEG channels.

	No. of Animals in the Cohort	No. of Seizures in the Left EEG Channel	No. of Seizures in the Right EEG Channel	No. of Non-Seizures in the Left EEG Channel	No. of Non-Seizures in the Right EEG Channel
HI–normothermia terms	7	470	538	2213	2212
HI–hypothermia terms	14	594	709	4423	4437
Sham–normothermia terms	5	0	0	2438	2457
HI–normothermia preterms	14	844	800	4443	4437
Sum	40	1908	2047	13,517	13,543
Total		3955		27,060	

**Table 4 bioengineering-11-00217-t004:** Results of the WS-CNN pattern classifier for the identification of post-HI seizures (HASs) across all study categories (#1–#3).

	TP Hits	FP Hits	FN Hits	TN Hits	Sensitivity [%]	Selectivity [%]	Precision [%]	Accuracy [%]	AUC	Average Accuracy [%]	Average AUC
**Study #1** 3 schemes	1636	8	29	8851	98.26	99.91	99.51	99.65	0.9959	98.48 ±1.01	0.9532 ±0.0403
	1044	259	27	8833	97.48	97.15	80.12	97.19	0.8991		
	938	70	5	4420	99.47	98.44	93.06	98.62	0.9647		
**Study #2** 3 schemes	1623	21	15	8865	99.08	99.76	98.72	99.66	0.9928	98.56 ±0.92	0.9530 ±0.0382
	1048	255	9	8851	99.15	97.20	80.43	97.40	0.9015		
	938	70	5	4420	99.47	98.44	93.06	98.62	0.9647		
**Study #3** 5 folds	780	11	2	5410	99.74	99.80	98.61	99.79	0.9929	99.78 ±0.04	0.9950 ±0.0013
	783	8	10	5402	98.74	99.85	98.99	99.71	0.9940		
	786	5	7	5405	99.12	99.91	99.37	99.81	0.9962		
	786	5	7	5405	99.12	99.91	99.37	99.81	0.9962		
	785	6	7	5405	99.12	99.89	99.24	99.79	0.9956		

**Table 5 bioengineering-11-00217-t005:** Results of the WF-CNN pattern classifier for the identification of post-HI seizures (HASs) across all study categories (#1–#3).

	TP Hits	FP Hits	FN Hits	TN Hits	Sensitivity [%]	Selectivity [%]	Precision [%]	Accuracy [%]	AUC	Average Accuracy [%]	Average AUC
**Study #1** 3 schemes	1628	16	7	8873	99.57	99.82	99.03	99.78	0.9947	98.50 ±1.28	0.9507 ±0.0444
	1030	273	57	8803	94.76	96.99	79.05	96.75	0.892		
	958	50	6	4419	99.38	98.88	95.04	98.97	0.9745		
**Study #2** 3 schemes	1615	29	5	8875	99.69	99.67	98.24	99.68	0.9909	98.10 ±1.36	0.9460 ±0.0402
	1041	262	109	8751	90.52	97.09	79.89	96.35	0.8933		
	915	93	1	4424	99.89	97.94	90.77	98.27	0.9538		
**Study #3** 5 folds	773	18	6	5406	99.23	99.67	97.72	99.61	0.9881	99.73 ±0.08	0.9920 ±0.0026
	781	10	6	5406	99.24	99.82	98.74	99.74	0.9931		
	780	11	7	5405	99.11	99.80	98.61	99.71	0.9924		
	785	6	3	5409	99.62	99.89	99.24	99.85	0.9959		
	776	15	1	5411	99.87	99.72	98.10	99.74	0.9904		

**Table 6 bioengineering-11-00217-t006:** Results of the 1D-CNN pattern classifier for the identification of post-HI seizures (HASs) across all study categories (#1–#3).

	TP hits	FP hits	FN hits	TN hits	Sensitivity [%]	Selectivity [%]	Precision [%]	Accuracy [%]	AUC	Average Accuracy [%]	Average AUC
**Study #1** 3 schemes	1610	34	70	8810	95.83	99.62	97.93	99.01	0.9857	98.00 ±0.96	0.9459 ±0.0364
	1048	255	79	8781	92.99	97.18	80.43	96.71	0.8977		
	916	92	1	4424	99.89	97.96	90.87	98.29	0.9543		
**Study #2** 3 schemes	1554	90	221	8659	87.55	98.97	94.53	97.04	0.9602	97.62 ±0.88	0.9426 ±0.0315
	1046	257	53	8807	95.18	97.16	80.28	96.95	0.8984		
	946	62	0	4425	100.00	98.62	93.85	98.86	0.9692		
**Study #3** 5 folds	782	9	9	5403	98.86	99.83	98.86	99.71	0.9935	99.70 ±0.14	0.9939 ±0.0029
	776	15	14	5398	98.23	99.72	98.10	99.53	0.9892		
	783	8	19	5393	97.63	99.85	98.99	99.56	0.9932		
	785	6	7	5405	99.12	99.89	99.24	99.79	0.9956		
	788	3	3	5409	99.62	99.94	99.62	99.90	0.9978		

**Table 7 bioengineering-11-00217-t007:** A comparison of the evaluated average overall performance of the proposed seizure detectors in each scheme of each study category.

	Study #1			Study #2			Study #3				
	Scheme 1	Scheme 2	Scheme 3	Scheme 1	Scheme 2	Scheme 3	Fold_1	Fold_2	Fold_3	Fold_4	Fold_5
Average overall performance from all classifiers (%)	99.48 ±0.34	96.88 ±0.21	98.63 ±0.28	98.79 ±1.24	96.90 ±0.43	98.58 ±0.24	99.70 ±0.07	99.66 ±0.09	99.69 ±0.10	99.82 ±0.03	99.81 ±0.07

## Data Availability

The data that support the findings of this study could be available from the corresponding author, H.A. or L.B., upon reasonable request.

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
