# Peer review of "Deep Learning for Generalized EEG Seizure Detection after Hypoxia–Ischemia—Preclinical Validation"

_bioengineering, 2024, doi:10.3390/bioengineering11030217_

Round 1

Reviewer 1 Report

Comments and Suggestions for Authors

This is a very nice, well written paper on a topic of clinical importance. My points for comment are relatively minor. 

1. Could the authors provide a little more detail on how the reference HAS were defined, e.g., was this done independently by more than one author, and were assessors blinded to experimental group?

2. Please define precision and accuracy as used in this context. 

3. How do EEG traces recorded at the dural surface compare to those measured at the scalp? The HAS seen much more defined that one would typically see in clinical practice. 

4. y-axes have variable scale and magnification. 

Author Response

Please review our responses in the attached file.

Reviewer 2 Report

Comments and Suggestions for Authors

The aim of this study was to evaluate the effectiveness of convolutional neural network-based deep learning classifiers in detecting hypoxic ischemic seizures in fetal sheep and to understand how maturity and brain cooling affect accuracy. Over 17,300 hours were recorded in the study using various cohorts, including normothermia and hypothermia periods. Seizure detectors trained with pre-normothermia data showed high accuracy (99.5%, 99.2 AUC), while those trained with normothermia and hypothermia data showed lower performance (98.6% accuracy, 96.5% AUC and 96.9% accuracy, 89.6 AUC). Despite the differences in spectral features, the detectors achieved an average accuracy of 99.7% (99.4 AUC) with 5-fold cross-validation. These findings highlight the reliability of the proposed deep learning algorithms in identifying post-HI seizures that exceed maturity and show minimal effect from hypothermia in 256Hz recordings. Conclusion;

1- The study is very detailed and contains features that will contribute to the literature.

2- The motivation and mathematical background of the study are satisfactory. 

However

3- The discussion section of the study is written in great detail, but the conclusion section should be written separately and the results should be emphasized. 

4- There are too many references to authors in the study, this selfcite should be reduced.

Author Response

(The authors gave the same response as above.)
